# Global Occurrence of Clinically Relevant Hepatitis B Virus Variants as Found by Analysis of Publicly Available Sequencing Data

**DOI:** 10.3390/v12111344

**Published:** 2020-11-23

**Authors:** Stoyan Velkov, Ulrike Protzer, Thomas Michler

**Affiliations:** 1Institute of Virology, Technical University of Munich/Helmholtz Zentrum München, Trogerstrasse 30, D-81675 München, Germany; stoyan.velkov@tum.de (S.V.); protzer@tum.de (U.P.); 2German Center for Infection Research (DZIF), Munich Partner Site, D-81675 Munich, Germany

**Keywords:** hepatitis B virus, genotype, serotype, escape mutation, pre-core mutation, nucleoside resistance mutation

## Abstract

Several viral factors impact the natural course of hepatitis B virus (HBV) infection, the sensitivity of diagnostic tests, or treatment response to interferon-α and nucleos(t)ide analogues. These factors include the viral genotype and serotype but also mutations affecting the HBV surface antigen, basal core promoter/pre-core region, or reverse transcriptase. However, a comprehensive overview of the distribution of HBV variants between HBV genotypes or different geographical locations is lacking. To address this, we performed an in silico analysis of publicly available HBV full-length genome sequences. We found that not only the serotype frequency but also the majority of clinically relevant mutations are primarily associated with specific genotypes. Distinct mutations enriched in certain world regions are not explained by the local genotype distribution. Two HBV variants previously identified to confer resistance to the nucleotide analogue tenofovir in vitro were not identified, questioning their translational relevance. In summary, our work elucidates the differences in the clinical manifestation of HBV infection observed between genotypes and geographical locations and furthermore helps identify suitable diagnostic tests and therapies.

## 1. Introduction

Hepatitis B is a major global health burden with nearly a quarter of the human population exposed to infection with hepatitis B virus (HBV), which is the causative agent [1]. While acute infection is self-limiting, it can cause symptomatic hepatitis and, in some cases, liver failure and death. In contrast, patients who develop chronic infection run the risk of developing long-term sequalae such as liver cirrhosis or hepatocellular carcinoma (HCC). A total of 257 million, or 3.5% of the world’s population, are estimated to be chronically infected [2]. HBV is a major cause of liver cirrhosis and one of the most prevalent carcinogens in the world [3,4]. While deaths due to other major pathogens such as human immunodeficiency virus (HIV), mycobacterium tuberculosis, and malaria are declining, HBV-related deaths—currently estimated at 887,000 per year [2]—are increasing, making HBV a leading cause of death attributed to infectious disease [5].

The spectrum of outcomes following HBV infection varies, and several factors are associated with certain phenotypes. These include age at the time of infection, immune status, human leukocyte antigen type [6], and ethnicity [7]. However, environmental factors, such as alcohol [8] or aflatoxin [9], also play a role in disease progression and incidence of HCC.

Additionally, several viral factors, including the viral inoculum size at infection [10], the viral genotype and serotype, as well as mutations ascribing the virus to a certain phenotype, have shown clinical relevance (Reviewed in [11]). HBV is classified into at least nine genotypes (A–I) that impact the primary transmission route, rate of progression after infection, response to interferon alpha treatment, and incidence of HCC [12,13]. A putative 10th genotype ‘J’ has been proposed [14]; however, as it has only been isolated from a single patient and displays greater homology to gibbon HBV, it is not universally accepted as an independent genotype [15].

HBV variants cannot only be differentiated by their nucleotide sequence but also their reactivity toward reference antibodies, which defines the “serotype”. Despite playing a relative minor role in current patient care, they can potentially impact the efficacy of vaccines or antibody-based therapies as well as the recognition by diagnostic tests. In addition to genotype and serotype, several mutations have been identified to alter the clinical outcome, diagnostics, and treatment response to HBV infection (reviewed in [16]). This includes mutations in the HBV surface antigen (HBsAg), which render the virus undetectable in diagnostic tests or evasion from vaccine-induced or therapeutic antibodies [17], and mutations in the reverse-transcriptase (RT) domain of the viral polymerase driving resistance to nucleoside treatment [18]. Finally, mutations in the basal core promoter (BCP)/pre-core region are associated with increased risk of fulminant hepatitis or HCC [19,20].

Several publications describe or review the clinical relevance of HBV variants [21,22,23,24,25,26,27,28,29]. However, whilst previous studies analyze the frequency of mutations in a certain genomic region, within selected populations or within specific genotypes [30,31,32,33,34], there is currently no comprehensive overview of the distribution of clinically relevant HBV variants between world regions or genotypes. However, this information could be useful to understand varying phenotypes of the different HBV genotypes and inform further optimization of diagnostic tests and treatment regimens. Therefore, we performed a computerized analysis to study the frequency of clinically relevant HBV variants within publicly available HBV sequences.

## 2. Materials and Methods

### 2.1. Retrieval of HBV Sequences

Hepatitis B virus (HBV) sequences were retrieved from the Nucleotide database (https://www.ncbi.nlm.nih.gov/nuccore) of the National Center for Biotechnology Information (NCBI). To identify the HBV sequences, the taxon ID 10407 that represents HBV in the NCBI taxonomy browser (https://www.ncbi.nlm.nih.gov/Taxonomy/Browser/wwwtax.cgi) was used. All available data for each entry were retrieved in GenBank format [35].

### 2.2. Allocation of Sequences to HBV Genotypes

Basic Local Alignment Search Tool (BLAST) [36] version 2.9.0+ was used to generate a reference nucleotide database to assign genotypes to the retrieved HBV sequences. The reference BLAST database was created from previously published sequences of full-length HBV genomes of each sub-genotype [37], which were combined with additional non-human HBV reference sequences. The non-human HBV references were added to exclude the possibility of mis-genotyping sequences that do not belong to human HBV. The default makeblastdb parameters for dbtype nucl were used with FASTA [38] format references retrieved from their GenBank entries for the BLAST database generation.

The list of accession numbers of human HBV genomes followed by sub-genotype used in the reference BLAST database are as follows: JN182318: A1; HE576989: A2; AB194951: A3; AY934764: A4; FJ692613: A5; GQ331047: A6; FN545833: A7; AB642091: B1; FJ899779: B2; GQ924617: B3; GQ924626: B4; GQ924640: B5; JN792893: B6; GQ358137: B7; GQ358147: B8; GQ358149: B9; AB697490: C1; GQ358158: C2; DQ089801: C3; HM011493: C4; EU410080: C5; EU670263: C6; GU721029: C7; AP011106: C8; AP011108: C9; AB540583: C10; AB554019: C11; AB554025: C12; AB644280: C13; AB644284: C14; AB644286: C15; AB644287: C16; GU456636: D1; GQ477452: D2; EU594434: D3; GQ922003: D4; GQ205377: D5; KF170740: D6; FJ904442: D7; FN594770: D8; JN664942: D9; FN594748: E; FJ709464: F1b; DQ899146: F2b; AY090459: F1a; DQ899142: F2a; AB036920: F3; AF223965: F4; GU563556: G; AB516393: H; FJ023659: I1; FJ023664: I2; AB486012: J.

The list of accession numbers of non-human HBV genomes followed by naming used in the reference BLAST database are as follows: K02715: GSHV; U29144: ASHV; AF193864: OGHBV; AJ251935: STHBV; AY226578: WMHBV; AY628097: WCHBV; JQ664503: GOHBV; JQ664509: CHHBV; KY962705: BATHBV; KC790373: BATHBV1; KC790374: BATHBV2; KC790375: BATHBV3; KC790376: BATHBV4; KC790377: BATHBV5; KC790378: BATHBV6; KC790379: BATHBV7; KC790380: BATHBV8; KC790381: BATHBV9; AB823662: GIHBV; KT893897: SGIHBV; KT345708: STHBV2; KY703886: CMHBV; MF471768: DHBV.

### 2.3. Analysis of Regional Frequency of HBV Genotypes and Clinically Relevant Variants

HBV sequences and associated data were downloaded and the information under FEATURES in the GenBank entry was evaluated for the country key. Countries were grouped in world regions as follows (number of included full-length HBV sequences for each country in brackets): Eastern Africa: Ethiopia (13), Kenya (17), Madagascar (1), Malawi (2), Mauritius (1), Rwanda (14), Somalia (9), Tanzania—United Republic of (3), Uganda (2), Zimbabwe (4); Middle Africa: Angola (14), Cameroon (62), Central African Republic (30), Congo—The Democratic Republic Of (5), Gabon (6); Northern Africa: Egypt (6), Sudan (17), Tunisia (5); Southern Africa: Botswana (10), Namibia (6), South Africa (58); Western Africa: Benin (4), Burkina Faso (17), Cape Verde (10), Gambia (2), Ghana (14), Guinea (74), Liberia (6), Mali (1), Niger (24), Nigeria (27); Caribbean: Cuba (8), Haiti (49), Martinique (23); Central America: Costa Rica (2), El Salvador (2), Mexico (27), Nicaragua (4), Panama (40); Northern America: Canada (54), Greenland (15), United States (508); South America: Argentina (131), Bolivia—Plurinational State of (11), Brazil (72), Chile (32), Colombia (1), Peru (3), Uruguay (8), Venezuela—Bolivarian Republic of (34); Central Asia: Kazakhstan (2), Tajikistan (8), Uzbekistan (10); Eastern Asia: China (2294), Hong Kong (75), Japan (281), Korea—Republic of (91), Mongolia (13), Taiwan—Republic Of China (57); South-Eastern Asia: Cambodia (28), Indonesia (120), Lao People’s Democratic Republic (43), Malaysia (195), Myanmar (18), Philippines (15), Thailand (107), Vietnam (145); Southern Asia: Bangladesh (81), India (330), Iran—Islamic Republic Of (53), Nepal (3), Pakistan (6); Western Asia: Saudi Arabia (4), Syrian Arab Republic (58), Turkey (83), United Arab Emirates (1); Eastern Europe: Belarus (8), Poland (33), Russian Federation (107); Northern Europe: Estonia (16), Ireland (1), Latvia (8), Sweden (15), United Kingdom (8); Southern Europe: Italy (47), Serbia (7), Spain (15); Western Europe: Belgium (116), France (12), Germany (12), Netherlands (6); Oceania: Australia (55), New Zealand (30), Fiji (5), New Caledonia (7), Papua New Guinea (16), Vanuatu (1), Kiribati (4), Samoa (1), Tonga (3).

### 2.4. Sequence Analysis

All sequences identified as non-human HBV with our BLAST database, as well as sequences classified as “unverified” or “non-functional” in the NCBI database were excluded. The occurrence of unclear bases in the sequence (residues labeled ‘*n*’) was an additional reason for exclusion. To allow analysis of sequences from different genotypes that vary in length, blank insertions were inserted into shorter sequences to achieve a length of 3257 bp. Sequences that did not start at the EcoRI site, which is generally considered as the start point for annotation, were corrected to allow for alignment with other sequences.

### 2.5. Analysis of Amino Acid Sequences

To analyze the HBV proteins, nucleotide sequences were translated to amino acid sequences with sixpack (EMBOSS) [39] using the orfminsize 100 and mstart options. Only open reading frames (ORF) starting with a methionine and a minimum length of 100 amino acids were taken into account. The correct reading frame of the respective protein was identified by a BLAST search against a database containing reference sequences of the HBV proteins. After assignment to the correct protein, a further size exclusion was performed by excluding implausible short proteins. Requirements for amino acid sequence length were ≥330 for pre-S1/pre-S2/S, ≥140 for pre-core/core, ≥700 for polymerase, and ≥100 for X. Proteins and nucleotide sequences were aligned in separate FASTA files using MUSCLE v3.8.1551 [40]. To account for the different lengths of the genotypes, sequences of each genotype were first aligned separately and, after including blank positions in the sequences of shorter genotypes, all sequences were combined into a single file.

### 2.6. Prediction of HBV Serotypes

Serotypes were predicted based on the amino acid variation at defined positions of HBsAg —122, 127, 134, 159, 160, 177, and 178—as previously described [41] and as outlined in Table 1. Sequences that could not be assigned to a specific serotype by the combinations of amino acids below were classified as undefined.

### 2.7. Analysis of Amino Acid Conservation and Frequency of Clinically Relevant Mutations

To analyze the conservation of nucleotide and protein sequences, each aligned FASTA file containing the full-length HBV genome sequence or the individual protein sequence was analyzed using a custom coded inhouse script. Briefly, each position of the analyzed sequence was individually assessed and the frequency of each nucleotide or amino acid counted. Based on the total number of sequences, the nucleotide/amino acid distribution was calculated for each position. The consensus sequence was generated by taking the most frequent nucleotide/amino acid at each position. Clinically relevant mutations were analyzed based on the overview established by Lazarevic et al. [42].

### 2.8. Data Processing

All data processing, including the evaluation of nucleotide and amino acid sequence conservation, serotype prediction, and analysis of frequency of clinically relevant mutations was performed with custom inhouse scripts written in Ruby programming language (https://www.ruby-lang.org). Graphical representations were created in Graphpad Prism 8.4.3 (https://www.graphpad.com/scientific-software/prism/). The phylogenetic tree was generated using RAxML-NG v. 0.9.0 [43] from the aligned HBV reference from the BLAST database. Visualization of the tree was performed with FigTree v1.4.4 (http://tree.bio.ed.ac.uk/software/figtree/).

## 3. Results

### 3.1. Sequence Acquisition and Processing

We retrieved 115,955 HBV nucleotide sequences (Figure 1A) from the NCBI database. Sequences that contained unspecified nucleotides or were marked as “non-functional” or “unverified” in the database entry were excluded. To allocate sequences to genotypes, a search was performed using BLAST against a database of reference *hepadnaviridae* genomes containing reference sequences of the different HBV genotypes and sub-genotypes as well as viruses with non-human hosts (see methods). After exclusion of non-human HBV sequences, 82,813 sequences were processed further.

Once the sequences were filtered by length, we selected only full-length genomes for further analysis. The size range was chosen to ensure that sequences of the shortest (genotype D with 3182 bp) as well as longest genotype (genotype A with 3221 bp) were represented. However, to account for the size distribution of available sequences and to allow the inclusion of variants with unusual length, the size range was enlarged from 3150 to 3275 bp (Figure 1B). This resulted in a panel of 7278 HBV full-length genome sequences for further analysis. Genotypes were differentially represented in our database. Genotype C had the most abundant full-length genome sequences (*n* = 2700) followed by genotype B (*n* = 1539), D (*n* = 1218), A (*n* = 1004), E (*n* = 312), F (*n* = 275), I (*n* = 96), G (*n* = 89), and genotype H (*n* = 44) (Figure 1C). When compared to the global genotype distribution as estimated by us previously [44], we found that genotype E sequences were strongly underrepresented (4.2% vs. 17.6%). Genotypes A and D were also slightly underrepresented (13.8% vs. 16.9%; 16.7% vs. 22.1%), whereas the remaining genotypes (B, C, F, G, H, I) were overrepresented in our database.

We also determined the origin of 6142 out of 7278 samples from the NCBI database. Sequences from countries within the same geographical area (for the definition of the geographical areas, see methods) were pooled for a better overview and to allow a more reliable calculation of the frequencies of clinically relevant HBV variants. The majority of samples originated from Eastern Asia (45.8% of samples, 34.4% alone from China), followed by South-Eastern Asia (10.9%), Northern America (9.4%), and Southern Asia (7.7%; Figure 1D). We noted a distinct distribution the HBV genotypes across the globe [44], with genotype B and C sequences mostly derived from East Asia and Southeast Asia, genotype D sequences from Southern Asia, Western Asia, and Europe, and genotype E sequences from Sub-Saharan Africa. Genotype A was widely distributed throughout world regions, with most sequences (302) originating in Northern America. In line with their geographical occurrence, genotype F, G, and H sequences were mainly identified in Southern America and genotype I sequences were mainly identified in South-Eastern Asia (Figure 1D). From the 7278 full-length nucleotide sequences, we predicted the amino acid sequences expressed from open reading frames. As some sequences did not contain start codons for certain ORFs, or coded for implausible short sequences due to premature stop codons, this resulted in 7211 pre-core/core, 7158 polymerase, 7157 pre-S1/pre-S2/S, and 7242 X protein sequences.

### 3.2. Distribution of HBV Serotypes

The structural basis that determines HBV serotype has been thoroughly studied, leading to the discovery that certain amino acids at positions 122, 127, 134, 159, 160, 177, and 178 of HBsAg (shown in Table 1) determine reactivity toward serological reference antibodies. This allows the serotype of an HBV variant to be predicted, depending on a known amino acid sequence [41]. Interestingly, when analyzing the conservation of HBsAg, we found significant variation especially at positions that determine the serotype (Figure 2A). The different amino acids found at these positions were mostly consistent with the algorithm determining the HBV serotypes (Table 1), as almost all sequences studied could be allocated to a certain serotype. Genotypes were each associated with a distinct serotype (Figure 2B). Most genotypes presented primarily with the adw serotype; however, the majority of the genotype A, I, G, and B sequences presented as adw2 and the genotypes H and F presented as adw4. In contrast, genotypes D and E presented with the ayw serotype, with genotype D almost equally divided into ayw2 and ayw3 serotypes, whereas genotype E was almost exclusively ayw4. In contrast, genotype C was the only genotype predicted to have the adrq+ serotype in significant quantities. When analyzing all HBV full-length sequences irrespective of genotype (pie chart labeled “Total”, Figure 2B) the majority of sequences had an adrq+ (30.9%, which derived almost exclusively from genotype C sequences, Figure 2C) or adw2 serotype (33.9%, mainly genotype B and A, Figure 2C), followed by ayw2 (11.0%) and ayw3 serotypes (7.8%). Although representing almost all genotype H and F sequences, the adw4q- serotype constituted only 4.2% of total sequences, reflecting the low number of included genotype H and F sequences (Figure 2C), due to the relatively low global occurrence [44].

### 3.3. Clinically Relevant HBsAg Mutations

Then, we interrogated HBsAg coding sequences for previously described clinically relevant mutations, which have been shown to cause occult infection, false negative diagnostic tests, or escape from vaccine induced or passively administered antibodies. The overall frequencies of such mutations were low, and the different mutations were relatively evenly distributed between genotypes (Figure 3). The only exception constituted the P127H/L mutations, which are associated with occult HBV infection appearing to be the wild-type amino acid for genotypes E, F, and H (96.8%, 98.9% and 97.7%). In contrast, the R122P (associated with occult infection) was not identified in any sequence, questioning its clinical significance. S136P and C139R were also very infrequent (each 0.04%) and only identified in genotype C. When looking at the regional distribution of HBsAg mutations (Figure 3B), P127H/L showed a concentration in the African continent as well as Central and South America, where most sequences belonged to either genotype E, F, or H (Figure 1D). Interestingly, the V177A mutation did not show a significant enrichment in any genotypes; however, it was found in approximately 33% of sequences from Oceania (including Australia, New Zealand, Melanesia, and Polynesia).

### 3.4. Frequency of Resistance Mutations against Reverse-Transcriptase Inhibitors

Then, we analyzed the conservation of the reverse-transcriptase (RT) domain of the polymerase protein. As this region is targeted with standard-of-care therapy using nucleoside analogues, mutations rendering the virus resistant to such treatments are highly clinically relevant. The different domains (A–G) of the RT that are crucial for function were found to be highly conserved throughout all genotypes, whereas the other regions showed greater variation (Figure 4A).

In a next step, we determined the frequency of mutations that are known to cause resistance to nucleoside therapy. M204V/I and L180M were the most abundant mutations found that were present in sequences of all genotypes, and they are associated with resistance toward lamivudine, telbivudine, and entecavir. Genotype A showed the highest frequency of both of these mutations (L180M: 13.8% and M204V/I: 14.0%) followed by genotype G (8.1% and 17.2%). High frequencies (between 5 and 9%) were also found in genotypes C, D, and G but were found in a minority of genotype E and I sequences (<1.0% and 1.1%, respectively). A few mutations have been found to drive resistance toward tenofovir treatment. These include A194T, which was identified in HIV/HBV co-infected patients and was shown to mediate a partial resistance toward tenofovir [45]. Two other mutations have been associated with tenofovir resistance, P177G and F249A; however, these mutations were created by site-directed mutagenesis in vitro [46], and it is unclear if they are found in patient-derived circulating virus. Interestingly, none of the >7000 sequences analyzed by us contained either the P177G or F249A mutation, which questions their clinical relevance. In contrast, A194T was identified in several genotypes, albeit in very low numbers, with the exception of genotype H, where 4.6% of sequences harbored this mutation. While only six out of 15 resistance mutations we looked for were found in genotype H, all of these were present above 2%. In genotype E, similar mutations were found as in genotype H, but with overall lower frequencies (each at or below 0.7%).

When analyzing the geographical distribution of each mutation, we found the highest frequency, especially of the L180M and M204V/I mutations, in Northern America (mainly comprising genotype A sequences, Figure 1D), followed by Europe and Eastern/Southern Asia (Figure 4C). In contrast, lower numbers were found on the African and Asian continents, as well as Central and South America. 

### 3.5. Distribution of Mutations in the Basal Core Promoter and Pre-Core Region

The basal core promoter (BCP) and pre-core regions of the HBV genome play an important role in determining the pathogenicity of HBV, and variants have been associated with an increased risk of fulminant hepatitis or HCC. Therefore, we analyzed the nucleotide sequence of this region and found the direct repeat 1 (DR1) region of the BCP and the epsilon stem loop to have the highest conservation across all genotypes (Figure 5A). As previously described, we found that genotype G sequences had an insertion at nucleotide position 1906–1941 (not shown), making it the longest of all genotypes (3248 bp). When analyzing the occurrence of clinically relevant mutations, we found BCP and pre-core mutations in relatively high abundance in almost all genotypes (Figure 5B). For genotype G, many “mutations” (C1653, T1753C, A1762T, G1764A and G1896A) were found with such high frequencies (93.3–100%) that they should be regarded as wild-type. Of all mutations, A1762T and G1764A were found at the highest frequencies in other genotypes, with decreasing rates in genotypes C, A, I, F, H, D, B, and E.

The majority of mutations did not show a distinct geographical enrichment, barring a few exceptions. G1862T, which was frequently found in the Caribbean, and G1896A, which was identified in more than 50% of sequences from Western Asia and Southern Europe, show distinct geographical enrichment that cannot be explained by genotype G, as it is rarely found in these regions (Figure 1D).

## 4. Discussion

HBV infection presents with a diverse disease profile with variation seen in the primary transmission mode, rate of disease progression, symptomatic disease, occurrence of sequalae, diagnostics, and treatment response. In addition to environmental and host factors, HBV genomic variation has been shown to be associated with a certain disease phenotype. In this study, we analyzed publicly available full-length HBV sequences to get an overview over the frequencies of clinically relevant HBV variants worldwide.

Importantly, the results of our study should be taken with caution, as several confounding factors potentially influenced the frequencies of mutations identified. Most importantly, sequences isolated from patients exhibiting an abnormal disease phenotype might be preferentially sequenced; thus, one can assume an enrichment of clinically relevant mutations compared to naturally circulating variants. Second, it is not clear if these mutations are associated with a specific disease phenotype in all genotypes; thus, harboring a certain mutation does not necessarily mean that there is an increased risk associated with this finding.

Furthermore, genotypes were differentially represented in our database with varying numbers of sequences. Genotypes A–D had between 1004 and 2700 sequences for each genotype; however, less sequences were available for the remaining genotypes (44 to 312). While this in large part reflects the global distribution, as previously reported [44], genotypes E–I sequences were underrepresented in our database. The differences in genotype frequency within sequenced isolates could be due to different prevalence in high vs. low income countries, as sequencing, at least when performed with modern technologies, is associated with significant cost. However, the relatively few sequences available for certain genotypes questions the precision of calculated frequencies. This is supported by the observation that in genotypes for which fewer sequences were available, many mutations were not found at all. Thus, low frequency mutations were possibly overlooked in these minority genotypes. This was even more relevant for the analysis of the regional distribution of HBV variants, as for some regions, relatively few sequences (between 20 and 2811) were available.

Another important factor to consider is that the majority of sequences were likely determined using classical Sanger sequencing of nucleic acids extracted from patient sera. The sensitivity of this approach is limited, with only variants occurring at a frequency of >20% within a viral population reliably detected [47]. Deciding which mutations are located on the same viral genome is much harder to achieve using next-generation techniques that traditionally have much shorter reads. Recent advances in next-generation sequencing (NGS) technology utilize longer reads and overlapping sequences, enabling the identification of mutations located on the same viral genome with high confidence [48]. NGS techniques are able to achieve a high sequence depth, readily detecting low frequency variants within a viral population. However, one needs to carefully evaluate the clinical relevance of these rare populations, which may simply be artefacts of high precision sequencing technologies.

A further possible reason for bias stems from difficulties in validating the quality of sequences included in our study. While information on sequence quality was in most cases not available, we tried to account for this by excluding sequences with database entries such as “unverified” or “non-functional” or which contained unspecified nucleotides (which could derive from poor sequencing quality). However, we cannot exclude other possible biases during sampling, sequencing, or database entry. Thus, conclusions based on low-frequency mutations should not be over interpreted.

For the prediction of HBV serotype, at least some confounding factors can be assumed to be less important, as it is less likely that the serotype influenced the likelihood of a certain variant to be sequenced. However, it is important to state that we only performed an in silico prediction and did not test the reactivity toward reference antibodies. While the relationship between amino acids at the relevant positions and the respective serotype is well established, we cannot exclude that some sequences would not show the predicted reactivity. Our finding that most genotypes have a distinct serotype confirms earlier observations in this regard [49]. We found that roughly 1/3rd of global HBV sequences showed either an adw2 or adrq+ serotype, with the other serotypes constituting the remaining 1/3rd of sequences. While serotypes are currently only playing a minor role in patient care, the information on occurrence of each serotype could still be helpful when designing future prophylactic or therapeutic vaccines or when developing antibody-based therapies (including antibodies to be passively administered but also bi-specific antibody constructs or chimeric antigen receptor T cells).

When analyzing the frequency of HBsAg mutants, we found that the overall frequency of such mutations was relatively low with no significant enrichment in any genotype. The only exception constituted mutations at position 127 (P to H or L), which are associated with occult HBV infection and seemed to be the wild-type for genotypes E, F, and H. Importantly, amino acid position 127 of HBsAg is also one of the determinants for the HBV serotype, and variants with Leucine (L) at this position are found in the ayw4 and adw4 serotypes (see Table 1). We observed the V177A mutation which was quite evenly distributed throughout genotypes but showed a regional enrichment in Australia, New Zealand, Melanesia, Polynesia (together defined as Oceania). Interestingly, both the V177A and P127L mutations are determinants of the HBV serotype and together with positions 122 and 160 of HBsAg define the adrq-serotype. Thus, the finding that these “mutations” are associated with occult infection could also mean that diagnostic tests used in these studies had a deficiency in recognizing this serotype. However, we cannot disregard factors, such as ethnicity, in driving selection pressure leading to an enrichment of these variants in certain regions.

For nucleoside analogue resistance mutations, we found that M204V/I and L180M displayed the highest frequency by far, which is consistent with the fact that they mediate resistance to the majority of currently used nucleoside analogues for hepatitis B therapy, including lamivudine. The relatively low barrier to resistance associated with these mutations likely led to relatively high occurrences, especially in regions with broad access to antiviral therapy. Along this line, nucleoside analogue resistance mutations in general showed the highest frequency in high-income regions such as Northern America (especially L180M and M204V/I), Europe, and Eastern/Southern Asia. In contrast, lower frequencies were found on African continent, Central and South America and Asia. This could indicate that more patients had been under treatment in these regions, causing a selective pressure on the virus to generate resistance mutations.

Of further interest, three different mutations have been described to interfere with tenofovir susceptibility, yet only one of these was found in virus isolated from patients (A194T; [45]), whereas the other two were generated in vitro (P177G and F249A; [50]). While we were able to find the A194T mutation in several sequences of different genotypes, the P177G and F249A mutations were not identified in any of the more than 7000 sequences analyzed by us. Therefore, it is unlikely that they constitute clinically relevant variants, and we speculate that these mutations create fitness losses for the virus that outweigh the advantages of tenofovir resistance.

The most interesting observation regarding the BCP/pre-core region was that almost all genotype G sequences contained several mutations (C1653, T1753C, A1762T, G1764A, and G1896A) which are associated with HBeAg negativity and increased risk for fulminant hepatitis and/or HCC. Whilst the lack of HBeAg expression is well described for genotype G [51,52], the risk of HCC and fulminant hepatitis is not as clearly associated, and may be hard to elucidate, as genotype G is mostly found in co-infections with genotype A [53]. The finding that there was regional enrichment of BCP/pre-core mutations G1862T in the Caribbean and G1896A in Western Asia and Southern Europe was also of note, especially as few genotype G sequences were identified in these regions. However, relatively low number of sequences (Caribbean: 80; Western Asia: 146; Southern Europe: 69) were available for all three regions; thus, future studies should seek to investigate if the increased frequency of these mutations are genuine and more importantly whether an increased risk for fulminant hepatitis or HCC is conferred.

In summary, we present an overview of the frequency of clinically relevant HBV variants in publicly available HBV sequencing data. While the results should be interpreted with care as several confounding factors could potentially have influenced the frequency of individual mutations described here, our data should help to identify interesting research questions for future studies, as well as help to design and select suitable diagnostic tests and therapies.

## Figures and Tables

**Figure 1 viruses-12-01344-f001:**
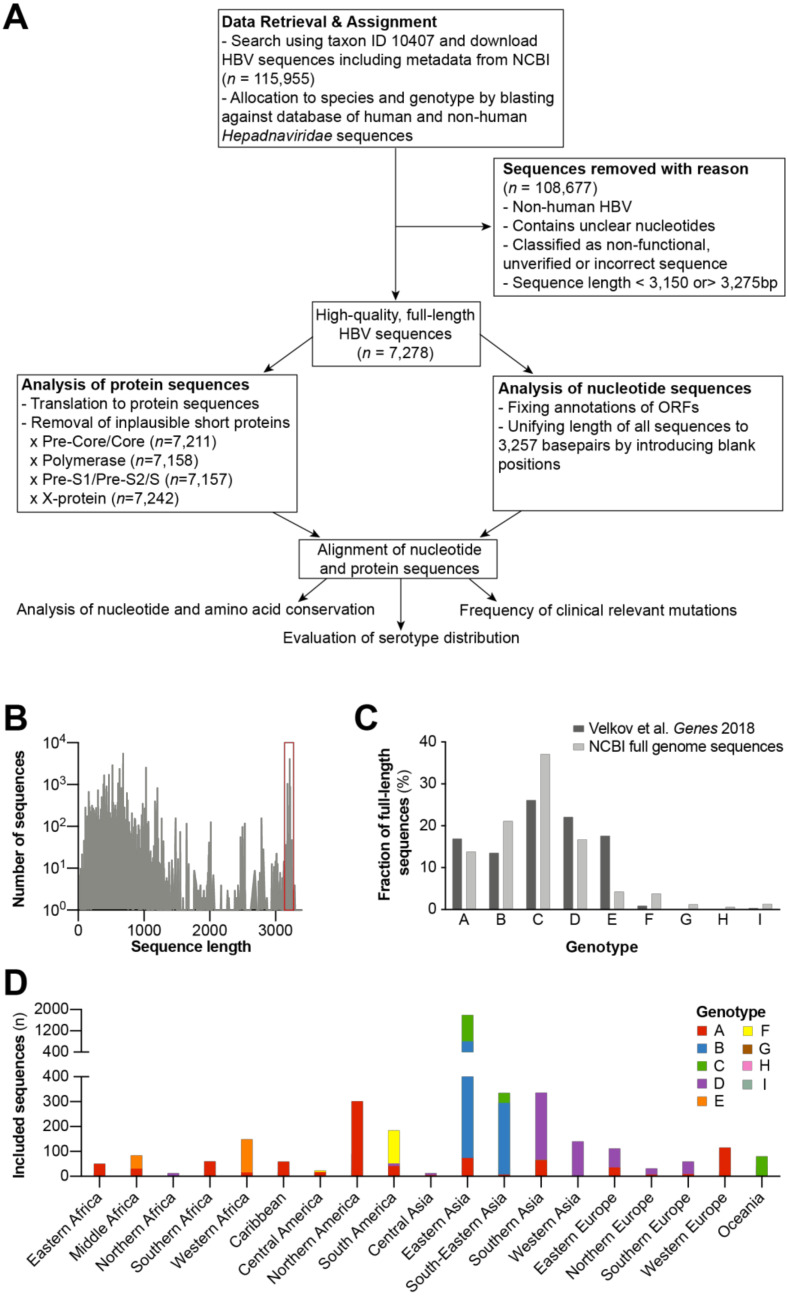
In silico analysis of publicly available full-length HBV sequences. (**A**) Work flow of study to estimate the frequency of clinically relevant HBV variants. (**B**) Retrieved HBV sequences were filtered by length, and only those between 3150 and 3275 base pairs (as indicated by the red box) were considered for further analysis. (**C**) Genotype distribution of included full-length HBV sequences compared to estimates of the global genotype distribution determined in Velkov et al. Genes 2018. (**D**) Overview of geographical origin and number of sequences for each genotype.

**Figure 2 viruses-12-01344-f002:**
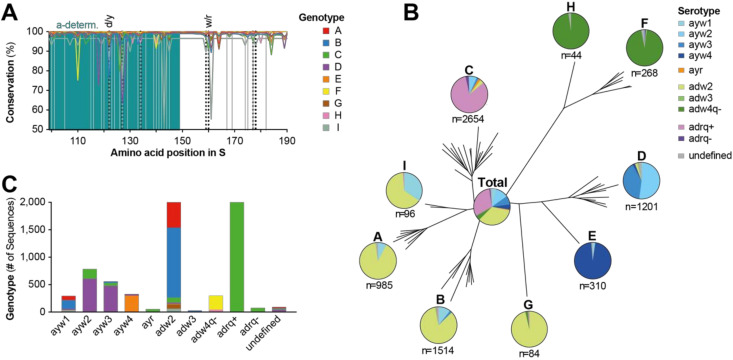
Distribution of HBV serotypes. (**A**) Amino acid conservation of HBsAg (starting at position 99 of S, marking the beginning of the major hydrophilic region) for each HBV genotype was determined by using the consensus sequence of each genotype as a reference. Relevant positions for serotype prediction are indicated by vertical dotted lines. Positions at which clinically relevant mutations occur (for details see Figure 3) are indicated by vertical gray lines. a-determ. = a-determinant. (**B**) Distribution of predicted serotypes within each HBV genotype. The serotype distribution of all global HBV sequences is shown in the middle. The phylogenetic tree was obtained using reference sequences for all HBV genotypes and sub-genotypes, which were employed to identify genotypes of sequences by Basic Local Alignment Search Tool (BLAST) search (see methods for details). (**C**) Total number of sequences with a certain serotype, subdivided by their genotype.

**Figure 3 viruses-12-01344-f003:**
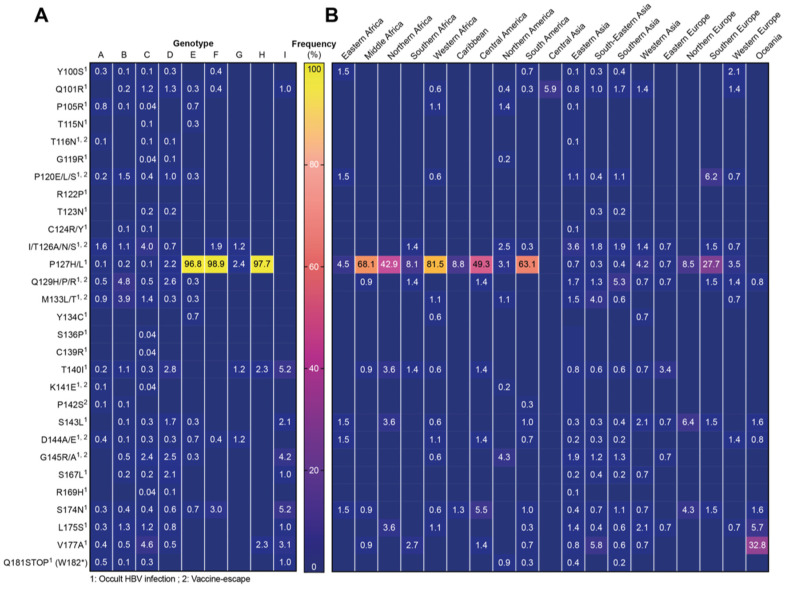
Frequency of clinically relevant HBsAg mutations. Numbers in the table indicate percentage of sequences containing the respective mutation within (**A**) each genotype or (**B**) different world regions. Blue fields without numbers represent a value of 0.0%. Superscript numbers indicate the clinical finding, which has been associated with the respective mutation.

**Figure 4 viruses-12-01344-f004:**
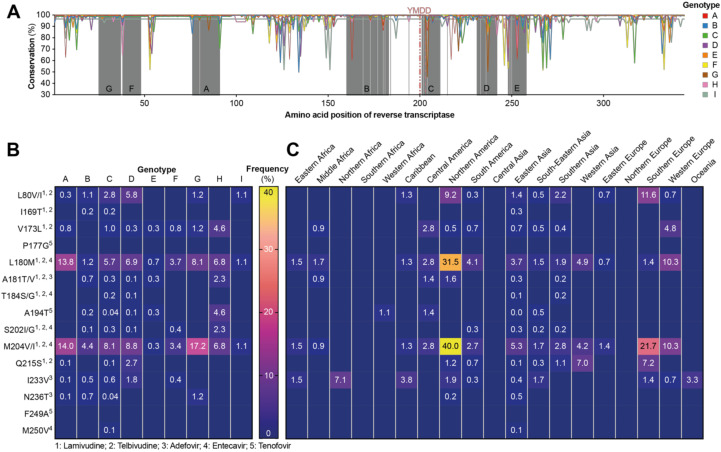
Frequency of mutations in the reverse transcriptase associated with resistance to nucleoside analogues. (**A**) Conservation of the reverse transcriptase (RT) part of the HBV polymerase within genotypes as determined using the consensus sequence of each genotype as a reference. The different domains (A–G) of the RT are highlighted by underlying gray color. Vertical light gray lines indicate positions at which resistance mutations occur. (**B**,**C**) Numbers in the table indicate percentage of sequences containing the respective mutation within (**B**) each genotype or (**C**) the different world regions. Blue fields without numbers represent a value of 0.0%. Superscript numbers indicate the clinical finding, which has been associated with the respective mutation.

**Figure 5 viruses-12-01344-f005:**
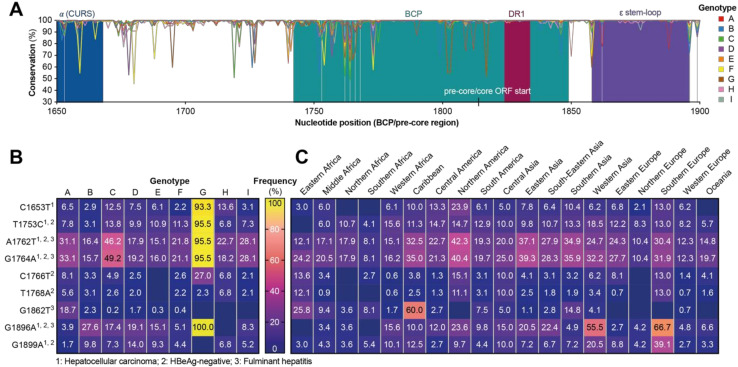
Frequency of clinically relevant mutations in basal core promoter or pre-core region of HBV. (**A**) Conservation of basal core promoter (BCP) and pre-core region of the HBV genome, vertical gray lines indicate positions as which clinically relevant mutations occur. Numbering according to convention starting at EcoRI site. α CURS = α core upstream regulatory sequence. (**B**,**C**) Frequency of clinically relevant mutations in the BCP/pre-core region within (**B**) each genotype or (**C**) the different world regions. Numbers in the table indicate percentage of sequences containing the respective mutation. Blue fields without numbers represent a value of 0.0%. Superscript numbers indicate the clinical finding, which has been associated with the respective mutation.

**Table 1 viruses-12-01344-t001:** Amino acid combinations within hepatitis B virus surface antigen (HBsAg) used to predict hepatitis B virus (HBV) serotypes. R = Arginine, K = Lysine, P = Proline, T = Threonine, L = Leucine, F = Phenylalanine, A = Alanine, V = Valine, Q = Glutamine.

Serotype	Amino Acid Position of HBsAg						
	122	127	134	159	160	177	178
ayw1 (option 1)	R	P	F	-	K	-	-
ayw1 (option 2)	R	P	-	A	K	-	-
ayw2	R	P	-	-	K	-	-
ayw3	R	T	-	-	K	-	-
ayw4	R	L	-	-	K	-	-
ayr	R	-	-	-	R	-	-
adw2	K	P	-	-	K	-	-
adw3	K	T	-	-	K	-	-
adw4q−	K	L	-	-	K	-	Q
adrq+	K	-	-	-	R	V	P
adrq−	K	-	-	-	R	A	-

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
