# Peer review of "Global Occurrence of Clinically Relevant Hepatitis B Virus Variants as Found by Analysis of Publicly Available Sequencing Data"

_viruses, 2020, doi:10.3390/v12111344_

Round 1
Reviewer 1 Report
Overall the paper presents interesting findings. However, it is very difficult to follow the paper due to the numerous grammatical mistakes. I have attached a pdf document with comments that provides representative comments which will help improve the quality of the paper.
Also, in Viruses, in 2018 a bioinformatics analysis based article was published - https://www.mdpi.com/1999-4915/10/11/603. Authors should take a look at it. This is just an example highlighting that similar studies have been performed, but authors have not cited many of them.

Reviewer 2 Report
In their manuscript entitled “Global occurrence of clinically relevant hepatitis B virus variants as found by analysis of publicly available sequencing data”, Velkov et al. have performed an in-silico analysis of publicly available HBV full-length genome sequences. The manuscript concerns the factors influencing the sensitivity of diagnostic tests for human hepatitis B viral infection, and the questions raised represent acute problems of modern health care.
The authors have written a high-quality manuscript. The data, available in NCBI, have been thoroughly analyzed, and the results obtained were clearly presented. The available data were filtered, and only full-length genomes included into further analysis.
At the same time, the authors do not comment on the quality of the original data in any way: whether or not all the data can have the same relevance in the analysis.
To date, NGS technologies, which allow one to obtain information about long sequences, are already available – for instance, Oxford Nanopore sequencing technology. The manuscript will benefit from including a discussion on whether such data can be compared with those obtained, for instance, using Illumina sequencing technology?
The manuscript is written in a good language. Nevertheless, the authors are encouraged to perform an additional spell check, just in order to eliminate several confusing misprints.
Minor comments:
- The authors are encouraged to define all the abbreviations used upon their first mention. Some abbreviations used in the manuscript – for instance, IFN (L. 11), HIV, TB (L. 34), HBsAg (L. 50) ‑ are not defined.
- L. 22-23: Written: “In summary, our helps explain …” Expected: “In summary, our present study helps explain …” or “In summary, our results help explain …” The sentence seems to be incomplete.
- P. 66-67: Written: “All available data for each entry was retrieved…” Expected: “All available data for each entry were retrieved…”
- The authors are kindly encouraged to carefully use commas, to make large sequences better understandable. In this way, for instance, the sequence on L. 65-66 could be re-written as follows: “To identify the HBV sequences, the taxon ID 10407 that represents HBV in the Taxonomy Browser (https://www.ncbi.nlm.nih.gov/Taxonomy/Browser/wwwtax.cgi) of NCBI, was used.”
The same regards the sequence on L. 115-116: “Sequences, which did not start at EcoRI site, were furthermore corrected to allow alignment with other sequences.”
- L. 111: Written: “ …as wells as…”. Expected: “…as well as…”
- L. 140: “analysed”; at the same time, on L. 141, the same word is written differently: “analyzed”. The authors are encouraged to unify the spelling throughout the manuscript.
- Figure 1, caption (L. 194-195): Written: “Retrieved HBV sequences were filtered for length and only sequences between 3,150 and 3,275 base pairs (as indicated by red box) considered for further analysis.” Expected: “Retrieved HBV sequences were filtered by length, and only those between 3,150 and 3,275 base pairs (as indicated by the red box) were considered for further analysis.”
- Figure 2, caption (L. 228-229): Written: “Phylogenetic tree was created with references sequences for all HBV genotypes and sub-genotypes which were used to identify…” Expected: “The phylogenetic tree was obtained using reference sequences for all HBV genotypes and sub-genotypes, which were employed to identify…”
- Figure 3, caption: L.247: Written: “Numbers in table indicate…” Expected: “Numbers in the table indicate…”
- 249-250: Written: “Blue fields without number represents a value of 0.0%. Superscript numbers indicate clinical 249 finding which has been associated with the respective mutation.” Expected: “Blue fields without numbers represent a value of 0.0%. Superscript numbers indicate the clinical finding, which has been associated with the respective mutation.”
I recommend minor revision. Just the points listed above should be addressed before it can be accepted.

Round 2
Reviewer 1 Report
I am happy with the revisions.